# Assessment of the Effect of A-PRF Application during the Surgical Extraction of Third Molars on Healing and the Concentration of C-Reactive Protein

**DOI:** 10.3390/pharmaceutics13091471

**Published:** 2021-09-15

**Authors:** Jacek M. Nowak, Stanisław Surma, Monika Romańczyk, Andrzej Wojtowicz, Krzysztof J. Filipiak, Maciej R. Czerniuk

**Affiliations:** 1Department of Dental Surgery, Medical University of Warsaw, 02-091 Warsaw, Poland; andrzej.wojtowicz@wum.edu.pl (A.W.); maciej.czerniuk@wum.edu.pl (M.R.C.); 2Faculty of Medical Sciences in Katowice, Medical University of Silesia in Katowice, 40-752 Katowice, Poland; stasiu.surma@onet.eu (S.S.); monika.romanczyk@med.sum.edu.pl (M.R.); 3Maria Sklodowska-Curie Medical Academy in Warsaw, Pałac Lubomirskich, 00-136 Warsaw, Poland; krzysztof.filipiak@uczelniamedyczna.com.pl

**Keywords:** wound dressing, healing process, tissue engineering, A-PRF, CRP, third molar

## Abstract

Extraction procedures for mandibular third molars are performed all over the world every day. Local inflammation resulting from surgery, and the pain that patients experience, often make it impossible to take up daily life activities, such as work or sports. Growth and anti-inflammatory factors, located in the fibrin network, have a positive effect on tissue-healing processes and should also reduce local inflammation. Advanced platelet-rich fibrin (A-PRF) applied locally influences such processes as: angiogenesis, osteogenesis and collagenogenesis. It also affects mesenchymal cell lines and anti- and pro-inflammatory mediators. Due to the autologous origin of the material, their use in guide bone regeneration (GBR) is more and more widespread in dentistry. The results of previous studies indicate that the use of A-PRF in the treatment area significantly reduces postoperative pain, while the formation of edema is not affected. C-reactive protein (CRP), which is an acute phase protein, appears in the blood as a consequence of inflammation. Due to the dynamics of changes in concentration of CRP, it is a protein that is sufficiently sensitive and is used in studies to monitor the tissue healing process. The effect of A-PRF application on CRP concentrations, before and after surgery, has not been investigated yet. The study was conducted on 60 generally healthy patients. A faster decrease of CRP levels was shown in patients who used A-PRF after the procedure. Additionally, it accelerated healing and reduced the occurrence of a dry socket close to 0.

## 1. Introduction

Extraction procedures for mandibular third molars are performed all over the world every day. According to Rosa, 90% of the human population has third molars, of which approximately 33% have at least one impacted tooth [1]. However, the data vary depending on the geographic region and population in which the research was conducted. According to Ayrancia’s own research, and the data he analyzed, the percentage of impacted molars was estimated to be between 40.5 and 75.6% [2,3,4,5,6].

Local inflammation resulting from surgery, and the pain that patients experience often make it impossible to take up daily life activities, such as work or sports. In addition, there is also a psychological aspect, as patients with significant swelling of soft tissues avoid meeting people, so they often stay at home. The possibility of reducing unpleasant, subjective sensations is one of the goals of improving the operator’s technique, as well as the reason for the search for new solutions and the use of new methods and materials in dentistry, such as fibrin preparations. Due to the action of growth and anti-inflammatory factors located in the fibrin network, there is a positive effect on tissue healing processes, and there should also be reduced local inflammation [7,8,9].

Advanced platelet-rich fibrin (A-PRF) is a second-generation blood derivative. It is distinguished from platelet-rich plasma by the formation of a fibrin scaffold in which there are stabilized platelets, which result in a slower release of growth factors [10,11,12]. The released factors include: VEGF (vascular endothelial growth factor), PDGF-AB (platelet derived growth factor AB), TGFβ-1 (transforming growth factor β-1), TSP-1 (trombospondin-1), IGF-I, IGF-II (insulin-like growth factors), EGF (epidermal growth factor), bFGF (basic fibroblast growth factor) [13,14].

Thanks to these factors, when A-PRF is applied locally, it influences such processes as: angiogenesis, osteogenesis and collagenogenesis. It also affects mesenchymal cell lines and anti- and pro-inflammatory mediators [15,16]. The results of previous studies indicate that the use of A-PRF in the treatment area significantly reduces postoperative pain, while the formation of edema is not affected [17].

Thanks to growth factors, we can consider this procedure as guided bone regeneration (GBR), i.e., a procedure in which an appropriate “scaffold” is created for the regeneration of bone tissue in the patient’s body. The purpose of the treatment is to support the regenerative processes of bone tissue and soft tissues [18]. It is extremely important, for clinical reasons, to minimize tissue loss after the extraction procedure. Thanks to this regeneration, it would be possible to insert an implant in the future. Due to the acceleration of regenerative processes, complications and hindering the healing processes occur less frequently [19,20,21].

Guided tissue regeneration is an ever-evolving field. Currently, many resorbable and nonresorbable materials are used in guided healing techniques.

The bone tissue in the masticatory system is extremely important. Bone tissue is constantly changing, as a result it can repair itself and adapt to new loads. The transformation of bone tissue involves the process of regeneration and modeling. Regeneration is the restoration of normal bone tissue at the site of its damage. Modeling, on the other hand, is the process of changing shape and size as a result of an acting stimulus.

Bone loss, as a result of tooth loss, is a significant clinical problem. Its possible consequences may be periodontal diseases of the neighboring teeth, and in extreme cases even jaw fracture.

Therefore, the healing of hard and soft tissues are considered an important step in treatment. As a result of studies at the microcellular level, platelets have been shown to play an extremely important role in wound healing. Patients undergoing extraction of the mandible third molar often experience severe pain, swelling and delayed healing. The healing process of the socket is composed of a number of biochemical, physiological and molecular sequences. These sequences are designed to restore tissue integrity and function.

Due to the possibility of side effects when using allogeneic materials, more and more hopes are being placed on autologous materials such as platelet-rich plasma and platelet-rich fibrin.

C-reactive protein (CRP), which is an acute phase protein, appears in the blood as a consequence of inflammation. It is produced under the influence of inflammatory cytokines in the liver, fat cells and arterial walls. Its concentration in the blood changes as a result of infection, inflammation, trauma, heart attack and during neoplastic diseases [22]. The CRP concentration is also influenced by sex, age, weight, the population that is being studied, medications taken, smoking and the method used for determination [23,24]. Due to the dynamics of changes in its concentration, it is a protein that is sufficiently sensitive and allows the monitoring of the tissue healing process [25,26,27,28]. Thanks to this, it is possible to use CRP as an indicator that is helpful not only in monitoring the course of treatment, but also in the early diagnosis of the inflammatory reaction [28,29,30,31]. The CRP concentration in a healthy person ranges from 0.1 to 3–9 mg/L depending on the method of determination. The increase in CRP concentration in the blood is noticeable 4–6 h after the injury, and its highest concentration is recorded after 24–72 h [26]. Its concentration increases twice every 8 h after surgery and returns to a normal concentration after 7 days [29]. In the case of severe injuries, infections with gram-negative bacteria, characteristic of inflammation in the oral cavity, as well as as a result of neoplastic processes, the protein concentration may increase 1000-fold, reaching a value >500 mg/L. CRP production in hepatocytes is induced by cytokines released during tissue damage and inflammation, in particular IL-6 (interleukin-6) and TNF-α (tumor necrosis factor-alpha) [27,29].

Studies have already shown a relationship between the surgical extraction of impacted molars and the concentration of CRP in the blood before and after the procedure [30]. The literature also includes data on CRP protein concentration in dental patients presenting with acute inflammation, periodontal abscess, submucosal abscess and dry socket [31,32]. In 2017, Graziani et al. conducted a study in which they characterized differences in biomarkers of systemic inflammation, vascular function and metabolism (that is, looking at highly sensitive C-reactive protein, lipids, fibrinogen, oxidative stress and analysis of endothelial function) in patients undergoing surgical extraction of the third molar [33]. However, the effect of A-PRF application on CRP concentrations before and after surgery has not been investigated yet.

## 2. Materials and Methods

Single-center prospective studies were conducted on patients of the Department of Dental Surgery, Medical University of Warsaw. All research procedures were carried out in accordance with the Helsinki Declaration of 1975, revised in 2013. The study was approved by the Bioethics Committee of the Medical University of Warsaw 8.10.2018 (approval number: KB/190/2018).

### 2.1. Experimental Groups

Sixty patients were randomly distributed between control and study groups. Their assignment to the study and control group was made randomly using a coin toss, where the reverse was the test group, and the obverse meant the control group. The coin toss was made by the patients themselves, who did not know the assignment of groups depending on the side of the coin (Figure 1).

### 2.2. Patient Selection

The inclusion criteria were designed to include patients between 18 and 40 years of age, in good general condition, requiring extraction of a partially impacted mandibular third molar. The study included only patients whose tooth needing extraction was in the mesio-angular position according to Winter and class II B according to Pell and Gregory (the crown of the third molar is covered by one-half of the front edge of the mandibular arm, the occlusal plane of the impacted molar tooth). The third molar is between the occlusal plane of the adjacent tooth and the neck of the adjacent tooth.

Patients were excluded from the study due to: poor oral hygiene, smoking, general diseases, diseases and treatment influencing blood biochemical parameters, pregnancy, breastfeeding, genetic diseases and craniofacial congenital malformations, people with advanced periodontal disease, people with multiple missing teeth, people with a BMI > 30, those taking anti-inflammatory drugs or if they had any other contraindications for oral surgery.

The clinical data of the patients are presented in the table. The gender distribution in the A-PRF group included 19 men (63.3%) and 11 women (36.6%). The mean age was 25.91 years for women and 25.42 for men. For the study group of 19 men (63.3%) and 11 women (36.6%), respectively, the mean age was 22.64 years for women and 25.58 for men (Table 1).

### 2.3. PRF Management

Before the procedure, blood was collected from a radial vein of those people in the study group using 4 A-X BY CHOUKROUN 10 mL tubes, then the original protocol proposed by Dr. Joseph Choukroun was applied. The blood was centrifuged in the PRF DUO QUATTRO centrifuge at 1500 rpm/for 14 min. After centrifugation, the obtained clot was collected from the tube, then separated from the plate mass and placed in a special PRF BOX (Figure 2). The two clots were left on a perforated clamp plate to drain and obtain the A-PRF membrane, the other two clots were placed in clamp containers for fibrin plugs.

### 2.4. Surgical Procedures and Intrasurgical Measurements

Before the procedure, blood was drawn from all patients to determine the concentration of CRP. Tooth extraction was performed under local anesthesia (Lignocainum 2% c. Noradrenalino 0.00125%). In all patients, the muco-periosteal envelope flap was incised and detached from the area around the teeth, 36–38 or 46–48. Then, an osteotomy was performed around the tooth crown using a rubella handpiece drill. The treatments were performed by the same operator. During the procedure, the time of the procedure was measured—all procedures were completed within 30–40 min. After the tooth extraction was performed in the study group, the previously prepared A-PRF preparation was placed in the alveolus in the form of two fibrin plugs and two membranes. For patients in the control group, placement of the preparation in the socket was simulated. Surgical wounds were fitted with single knotted sutures using Safil 3.0 absorbable synthetic sutures. Postoperatively, patients from both groups were treated with the same pharmacotherapy, 1 g of amoxicillin—1 tablet every 12 h—and non-steroidal anti-inflammatory painkillers (NSAIDs). Sutures were removed 7 days after surgery, and patients were re-tested for CRP concentrations.

### 2.5. Blood Test Methodology

The material for laboratory tests was venous blood, collected from a radial vein using the Biomedico collection kit (21GX3/4). Blood was collected in the morning between 8:00 and 10:00, after rest, in a sitting position, having eaten a light breakfast. BD Vacutainer SST TM II Advance 8.5 mL tubes with silica gel as a clot activator were used. The blood tubes were centrifuged according to the manufacturer’s recommendations, 1300–2000 *g* for 10 min. Patients’ blood was tested in the Diagnostic Laboratory of the Infant Jesus Hospital in Warsaw.

### 2.6. Methodology of Blood Biochemical Testing

The Hs-CRP test (high sensitivity C-reactive protein) was used for the biochemical test–determination of the concentration of CRP in blood serum—a test of high sensitivity, detecting changes in the concentration of CRP protein in undiluted samples of human serum, plasma or homogenized tissues. It allows you to precisely determine even a low concentration of CRP. It is a quantitative sandwich ELISA (enzyme linked immunosorbent assay) test, i.e., a double binding test. It involves the binding of an antigen between two layers of antibodies. The sensitivity of this test is 0.1 µg/mL, and the detection range is 0.25 µg/mL to approximately 8–10 µg/mL. The concentration of C-reactive protein was determined immediately before the procedure (CRP I) and again 7 days after the procedure (CRP II). Additionally, the difference between the two measurements (∆ CRP) was also determined.

The rate of ∆CRP was evaluated as the difference between CRP I and CRP II and ex- pressed as a number using the formula: ∆CRP = CRP II − CRP I

Based on the clinical examination and subjective feelings of the patient, the occurrence of a complication, i.e., difficult healing in the form of a dry socket, was also assessed. The rate of ∆CRP was evaluated as the difference between CRP I and CRP II and expressed as a number using the formula:∆CRP = CRP II − CRP I(1)

### 2.7. Control Day

Seven days after the surgery, the sutures were removed and a short postoperative questionnaire was completed regarding the presence or absence of signs of local inflammation based on a clinical examination:swellingpainsubjective trismusredness at the treatment sitedry mouthburning

### 2.8. Statistical Analysis

The conducted studies were randomized, prospective and screening studies. During the research work, the results obtained in both women and men were analyzed. In the statistical analysis, same-sex groups were not distinguished. The obtained test results were subjected to statistical analysis, taking into account the parameters of descriptive statistics: mean values, standard deviations as well as minimum and maximum values for measurable variables in the test and control groups. In order to compare the parameters consistent with a normal distribution, the t-student test was used, the remaining data were statistically analyzed using non-parametric methods—U Mann–Whitney. Statistical significance was assumed as *p* < 0.05. Statistica 13.3 (StatSoft) was used for the analysis.

## 3. Results

### 3.1. CRP

The study looked at data on age, gender, CRP levels, and its effect on symptoms of inflammation. The results of individual studies in both groups are presented in Table 2.

The performed statistical analysis showed that the mean value of CRP II and the difference in CRP ∆ in the control group (2.98 ± 2.54 and 1.79 ± 2.15, respectively) were significantly higher than in the study group and this was statistically significant (1.66 ± 0.93 mg/L and 0.17 ± 1.7 mg/L) (Table 3).

### 3.2. Postoperative Survey

During the inspection, the doctor also assessed features that may indicate the presence of local inflammation. The results of clinical trials and their analysis are presented in Table 4; Table 5 and Figure 3.

On the basis of the obtained results, it was found that the difference in the occurrence of swelling between the groups was statistically significant. The average frequency of swelling in the study group was 0.1 ± 0.31, while in the control group it was 0.53 ± 0.51.

The difference in the occurrence of pain, trismus and redness on the seventh day, postoperative, was also statistically significant. In the study group, the above features were not found (0; 0; 0, respectively), and in the control group, they were 0.9 ± 0.3; 0.3 ± 0.47; 0.4 ± 0.5.

There were no statistically significant differences in the occurrence of dry mouth and burning in the mouth between the study and control groups.

During the follow-up examination, the incidence of post-extraction complication was also determined, i.e., local inflammation in the form of a dry socket (Table 6).

In patients from the control group, the occurrence of dry socket 0.5 ± 0.51 was significantly more frequent statistically compared to the study group 0.1 ± 0.31.

## 4. Discussion

The concentration of CRP protein as a marker of inflammation is closely related to the healing process of postoperative tissues. It is dependent on many factors, and blood levels are affected by inflammation. A standard test measures a much wider range of CRP concentrations, but is less sensitive in the lower ranges, and the high-sensitivity CRP (hs-CRP) test detects lower concentrations of protein more accurately (it is more sensitive), making it more useful than the standard CRP test in the evaluation of the treatment process. Depending on the type of high-sensitivity test used, there is a difference in accuracy, which has been confirmed by numerous comparative studies [34,35,36,37].

It takes about 7 days for CRP to return to normal concentration levels [29]. However, from the work of Ceiod, the increased concentration may persist for up to 2 months [38]. The author’s own research showed statistically significant differences in the tissue regeneration process between the study group and the control group. An analysis of the concentration of CRP protein before and 7 days after the procedure showed that in both groups of patients, its concentration was lower than 10 µg/mL, so it was within the normal range. However, the mean values of CRP II protein concentration indicated slightly better regenerative processes in the study group as compared to the control group.

The faster return to normal concentration levels of patients treated with A-PRF is a result of better tissue healing. A-PRF is a clot that not only releases growth factors, but also mechanically covers the bone tissue, isolating it from the oral cavity environment, and thus saliva, which contains millions of bacteria [39]. Thanks to its antibacterial effect, it is not possible to colonize the clot. However, Al-Hamed states that PRF has no effect on the infection of the socket (understood as a socket with purulent exudate), reddening of the area and increased body temperature [40]. Furthermore, these findings were replicated in our own research and that conducted by Fujioka-Kobayashi et al. A literature review showed that the use of A-PRF significantly reduced the percentage of dry socket [41]. Thanks to anti-inflammatory factors, the body’s response to an extraction injury is also reduced. By reducing the secretion of pro-inflammatory factors in the treatment area, the concentration of CRP protein in the blood is also lower. Moreover, the presence of white blood cells in A-PRF inhibits the growth of bacteria.

There have been numerous research papers describing the postoperative symptoms and healing process in patients undergoing surgical removal of the third molar in the mandible [42,43,44,45,46,47,48,49,50,51,52,53,54,55]. One of the causes of persistent pain after extraction of the third molar is the occurrence of post-extraction local inflammation in the form of a dry socket. It is the most common complication and its frequency is determined to be from 7% to 35% [56,57,58,59,60]. There are many publications on the etiology and techniques of dry socket treatment in the literature [60,61,62,63,64]. A dry socket is local inflammation of the bone resulting from a clot not covering the bone tissue. The pain is felt until the connective tissue is granulated and the bone tissue is “covered”. A dry socket is diagnosed as a continuous throbbing postoperative pain in the extraction area that is not adequately relieved by painkillers. The authors’ own research shows that in the patient study group, they developed a dry socket 40% less often than in the control group. Only 10% of patients in this group showed the features of a dry socket. A statistically significant difference between the two groups correlates with the results obtained by other researchers. According to Al-Hamed, A-PRF significantly reduced the incidence of a dry socket complication compared to the control group [40]. Hoaglin’s studies also document a reduction in the incidence of dry socket by 9.5% in the control group and 1% in the group with PRF [65]. As stated in the meta-analysis prepared by Al-Hamed, the most significant and reliable results are those obtained by Eshghpour [66]. However, Asutay’s research did not confirm a statistically significant difference between the groups [67]. The beneficial effects of using A-PRF result from the presence of the many growth factors in it. These factors stimulate cellular mitosis and differentiation, increase collagen production, recruit leukocytes and other cells to the surgical site and initiate vessel growth. This promotes the healing of soft and hard tissues and at the same time intensifies angiogenesis. Moreover, the presence of white blood cells in A-PRF inhibits the growth of bacteria. Similar results were obtained in 2007 by Rutkowski in his research on the use of PRP placed in the extraction socket [68].

Also noteworthy is the study by Ratiu et al., in which they assessed the properties of various commercially available resorbable collagen membranes in the guided regeneration of bones after the addition of growth factor-rich plasma (PRGF) [69]. The structural and morphological features of three different commercial collagen membranes were tested. It was found that features such as porosity, fiber density, and surface topography can influence the mechanical behavior and performance of the membranes. Using spectroscopy, it was shown that the collagen matrix can act as a natural environment for supplying growth factors. The mechanical properties of the membranes were tested before and after soaking in PRGF. Tests showed that PRGF-modified membranes degraded more slowly compared to native membranes.

## 5. Conclusions

It should be noted that the concentration of reactive protein C in the peripheral blood, 7 days after the surgical extraction of the impacted tooth, is lower in patients who received A-PRF blood product intra-operatively. The decrease in CRP concentration proves that dental problems affect the general health condition of patients. Even a slight inflammation associated with the difficult eruption of third wisdom teeth causes a slight increase in CRP protein. The A-PRF preparation, on the other hand, allows for a faster reduction of CRP concentration after the procedure. In addition, A-PRF reduces the occurrence of symptoms indicative of local inflammation and significantly reduces the incidence of postoperative complication in the form of a dry socket.

## Figures and Tables

**Figure 1 pharmaceutics-13-01471-f001:**
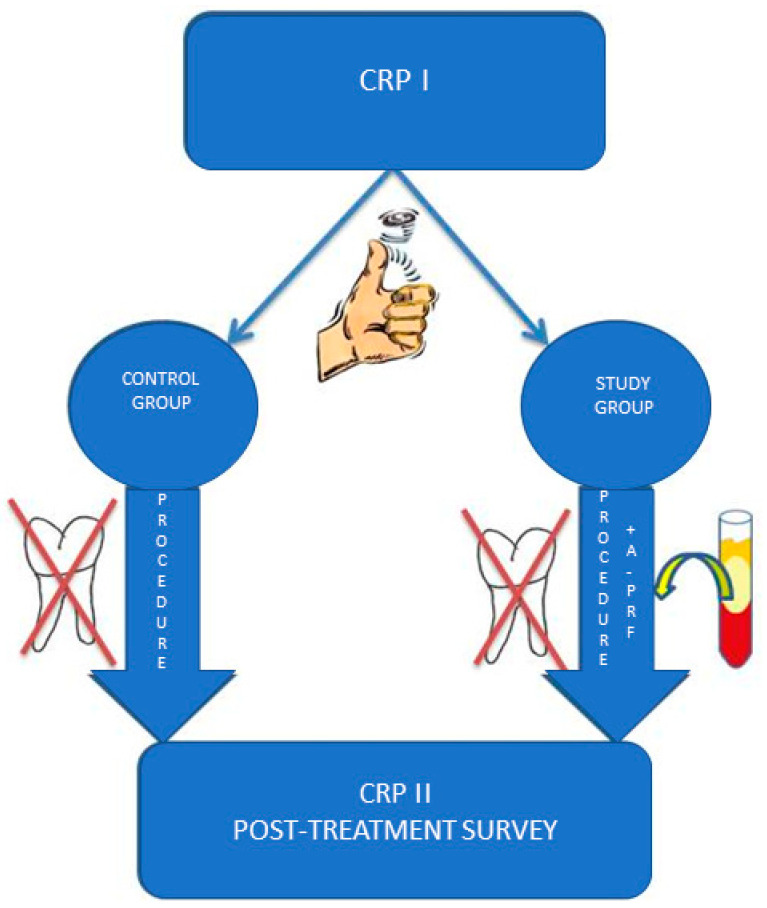
The scheme of the conducted study.

**Figure 2 pharmaceutics-13-01471-f002:**
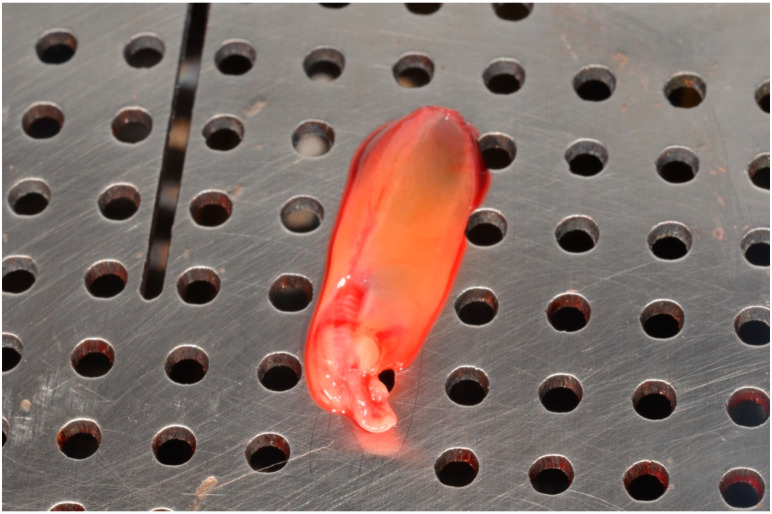
A-PRF clot on a perforated plate of the PRF-BOX system.

**Figure 3 pharmaceutics-13-01471-f003:**
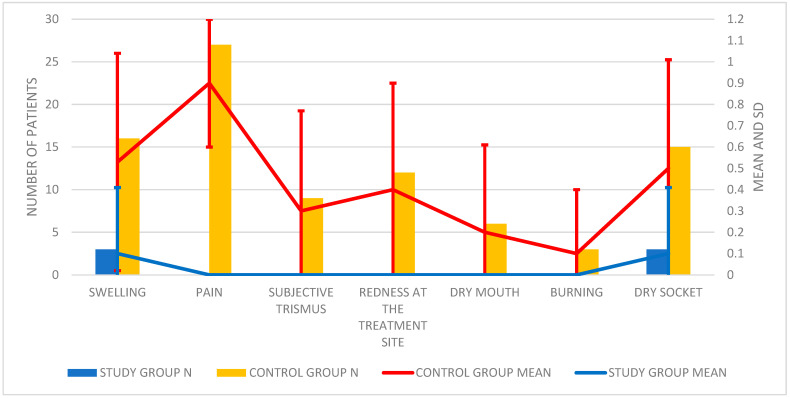
Histogram showing the number of symptoms of inflammation and complications in the form of a dry socket.

**Table 1 pharmaceutics-13-01471-t001:** Age, number and sex of patients in particular groups. Data are presented as abundance or mean ± standard deviation (SD); f–female; m–male.

Lables	Study Group	Control Group
number of participants	30	30
sex	f: 1; m: 19	f: 11; m: 19
age (mean)	f: 25.91 ± 5.61; m: 25.42 ± 3.55	f: 22.64 ± 2.46; m: 26.58 ± 3.80

**Table 2 pharmaceutics-13-01471-t002:** Data of the procedures: ∆CRP–difference in CRP concentration between CRP II and CRP I.

Group	Patient	Age	Sex	CRP I	CRP II	∆CRP
Study	1	26	M	0.40	0.20	−0.20
	2	20	M	0.44	2.68	2.24
	3	36	K	0.51	0.14	−0.37
	4	23	M	1.50	2.29	0.79
	5	26	M	3.26	1.38	−1.88
	6	36	K	1.72	1.99	0.27
	7	23	M	1.82	1.96	0.14
	8	22	K	1.30	1.14	−0.16
	9	26	M	2.10	0.90	−1.20
	10	27	K	4.98	1.50	−3.48
	11	26	M	0.30	1.14	0.84
	12	22	M	1.22	3.91	2.69
	13	27	K	0.19	2.20	2.01
	14	26	M	3.30	1.34	−1.96
	15	32	M	4.00	2.50	−1.50
	16	24	K	1.50	1.00	−0.50
	17	20	K	0.12	2.13	2.01
	18	22	K	1.50	1.10	−0.40
	19	20	K	0.15	2.10	1.95
	20	33	M	3.90	2.30	−1.60
	21	22	M	3.10	1.20	−1.90
	22	27	K	4.96	1.54	−3.42
	23	25	M	0.37	0.26	−0.11
	24	27	M	2.09	0.94	−1.15
	25	21	M	0.35	2.66	2.31
	26	31	M	4.19	2.54	−1.65
	27	24	K	1.25	1.05	−0.20
	28	26	M	1.30	1.10	−0.20
	29	23	M	1.15	3.80	2.65
	30	25	M	2.15	0.97	−1.18
Control	1	25	K	4.95	7.50	2.55
	2	34	M	0.50	4.70	4.20
	3	25	M	0.30	0.90	0.60
	4	27	M	2.30	1.70	−0.60
	5	26	K	1.34	6.30	4.96
	6	20	K	0.10	0.31	0.21
	7	26	M	0.75	2.10	1.35
	8	26	M	2.00	1.60	−0.40
	9	20	K	0.01	0.21	0.20
	10	23	K	0.34	8.17	7.83
	11	26	M	2.12	1.94	−0.18
	12	33	M	0.32	4.83	4.51
	13	37	M	0.60	5.03	4.43
	14	24	M	1.34	2.27	0.93
	15	25	K	4.95	7.50	2.55
	16	24	K	0.70	8.00	7.30
	17	26	M	0.50	2.00	1.50
	18	25	M	0.15	0.95	0.80
	19	25	M	0.50	1.30	0.80
	20	23	M	1.34	2.27	0.93
	21	21	K	1.51	2.49	0.98
	22	25	K	3.95	6.40	2.45
	23	25	M	0.20	0.90	0.70
	24	25	M	0.30	0.90	0.60
	25	25	M	0.29	0.80	0.51
	26	24	M	1.20	2.07	0.87
	27	26	M	0.45	2.15	1.70
	28	20	K	0.05	0.35	0.30
	29	23	M	2.35	3.25	0.90
	30	20	K	0.30	0.61	0.31

**Table 3 pharmaceutics-13-01471-t003:** Descriptive statistics of CRP I, CRP II and CRP ∆ protein concentration in the study and control groups. *n*–number of patients; SD–standard deviation, *p* TEST–power of test.

	Group	*n*	Mean	Median	Minimum	Maximum	SD	*p* TEST
CRPI	Study	30	1.84	1.5	0.12	4.98	1.48	0.083
	Control	30	1.19	0.55	0.01	4.95	1.35	
CRP II	Study	30	1.66	1.44	0.14	3.91	0.93	0.01
	Control	30	2.98	2.08	0.21	8.18	2.54	
CRP ∆	Study	30	0.17	−0.2	−3.48	2.69	1.70	0.001
	Control	30	1.79	0.915	0.60	7.83	2.15	

**Table 4 pharmaceutics-13-01471-t004:** Occurrence of symptoms of inflammation on day seven, postoperative. Data presented as the number and percentage of a given group.

Patients Who Have Occurred	Swelling		Pain		Subjective Trismus		Redness at the Treatment Site		Dry Mouth		Burning		Dry Socket	
	N	%	N	%	N	%	N	%	N	%	N	%	N	%
Study Group	3	10	0	0	0	0	0	0	0	0	0	0	3	10
Control Group	16	53.33%	27	90%	9	30%	12	40%	6	20%	3	10%	15	50%

**Table 5 pharmaceutics-13-01471-t005:** Descriptive statistics of inflammatory symptoms occurring on day seven, postoperative, in the study and control groups. *n*- number of patients; SD–standard deviation, *p* TEST–power of test.

	Group	*n*	Mean	Median	Minimum	Maximum	SD	*p* Test
Swelling	STUDY	30	0.1	0	0	1	0.31	0.004
	CONTROL	30	0.53	1	0	1	0.51	
Pain	STUDY	30	0	0	0	0	0	0.000
	CONTROL	30	0.9	1	0	1	0.30	
Subjective trismus	STUDY	30	0	0	0	0	0	0.047
	CONTROL	30	0.3	0	0	1	0.47	
Redness at the treatment side	STUDY	30	0	0	0	0	0	0.008
	CONTROL	30	0.4	0	0	1	0.50	
Dry mouth	STUDY	30	0	0	0	0	0	0.185768
	CONTROL	30	0.2	0	0	1	0.41	
Burning	STUDY	30	0	0	0	0	0	0.510598
	CONTROL	30	0.1	0	0	1	0.30	

**Table 6 pharmaceutics-13-01471-t006:** Descriptive statistics of the occurrence of a complication in the form of a dry socket on the day of the inspection in the study group and in the control group. *n*–number of patients; SD–standard deviation; *p* TEST–power of test.

	Group	*n*	Mean	Median	Minimum	Maximum	SD	*p* TEST
Dry Socket	Study	30	0.1	0	0	1	0.31	0.007959
	Control	30	0.5	0.5	0	1	0.51	

## Data Availability

Not applicable.

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
