# Peer review of "Assessment of the Effect of A-PRF Application during the Surgical Extraction of Third Molars on Healing and the Concentration of C-Reactive Protein"

_pharmaceutics, 2021, doi:10.3390/pharmaceutics13091471_

Round 1

Reviewer 1 Report

Artical name : Assessment of the effect of A-PRF application during the surgical extraction of third molars on healing 2 and the concentration of C-reactive protein.

This paper was discussing the influence of A-PRF locally application on the CPR content in a freshly extracted third molar. The results of previous studies indicate the significant reduction of postoperative pain and edema by using A-PRF but this was the first paper to investigate the effect of A-PRF application on CRP concentrations before and after surgery. as CPR protein concentration was presenting with acute inflammation, periodontal abscess, submucosal abscess, dry socket.

the paper has 2 groups control and study group and they use ELISA test for blood biochemical testing before the surgery and after 1 week of the surgery and the results shows a significant difference between the 2 groups in the CPR concentration with a decrease of inflammation signs and acceleration of healing process in the study group.

The topic was interesting but I have some points on the manuscript I would like the author to consider:

  1. in the materials and methods part, you say that CPR1 was CPR concentration before surgery, and CPR2 is the concentration 1 week after the surgery. But why the sample was collected next morning?
  2. Figure legend should be added to all figures.
  3. Pic2 is mentioned in the manuscript but not specified which one. would you mind marking the figure.
  4.  In 174 line you mentioned "fig 4", but figure 4 is not to be found in all the manuscript.

Reviewer 2 Report

The current paper presents a study regarding Advanced Platelet Rich Fibrin application in 60 patients requiring extraction of a partially impacted mandibular third molar, and assessment of C-reactive protein (CRP), which is one of the acute phase proteins, appears in the blood as a consequence of inflammation. The authors concluded that the concentration of reactive protein C in the peripheral blood 7 days after the surgical extraction of the impacted tooth is lower in patients who received A-PRF blood product intra-operatively. The A-PRF preparation, on the other hand, allows for a faster reduction of CRP concentration after the procedure. In addition, A-PRF reduces the occurrence of symptoms indicative of local inflammation, and significantly reduces the incidence of postoperative complication in the form of a dry socket.

Comments and observations.

Although the subject is very interesting and actual, it doesn’t match to the aim and scope of the journal. According to the journal information, covered topics include pharmaceutical formulation, drug delivery, pharmacokinetics, biopharmaceutics, pharmacogenetics, bio-engineering. So, I consider that the authors should submit their paper to a journal devoted to clinical dentistry topics, which would be more adequate.

However, I have some general comments:

  1. Preparation of Plasma Rich in Growth Factors (PRGF), Platelet Rich Fibrin or any other similar product, require the assessment of Hematology Parameters (Leukocytes, Erythrocytes , Platelets , etc…) and Growth Factor Content such as Quantitative assessment of the main growth factors, cytokines, and chemokines, etc. These parameters are missing from the results. For more details, I recommend a reference paper published by Ratiu et al. (2018) Appl. Sci. 2019, 9(5), 1035; https://doi.org/10.3390/app9051035, in which the complete characterization of the autologous formulation is presented.
  2. The introduction section should be reconsidered by highlighting the aim of the work, which is completely missing. These aspects have to be mentioned also in the abstract.
  3. The protocol and research design should be improved.
  4. The discussion section needs to be improved by adding references strictly related to subject, while removing some which are not relevant.
  5. The conclusion section does not present enough outputs.

My recommendation is rejection and re-submission to a more clinically-related journal.

Author Response

  Thank you very much for your extensive opinion and comments. We gave up on full blood tests because we wanted a group that reflected clinical reality. After training with Dr. Dhoukroun (the main publisher on PRF), we came to the conclusion that doing research on a carefully selected group will distort the real results. As a rule, the autologous preparation fulfills its task in the situation of generally understood health, in patients without general diseases. We appreciate your comments on the research structure and the article, we will certainly use them in the future to improve our work. We were very eager to read the indicated article. If we are able to edit the main text, we will gladly include it as one of the items in the bibliography.

Reviewer 3 Report

In this study, the A-PRF was used as a post-surgical treatment in order to a faster reduction of CRP concentration and reduction of the occurrence of symptoms indicative of local inflammation, and the incidence of postoperative complication in the form of a dry socket. The topic of study is interesting and novel but the manuscript requires minor revisions to be published.

  1. Figure captions are missing
  2. The order number of fig.3 and fig.4 should be changed.
  3. Fig 4 is missing
  4. Table 6 is missing.
  5. I suggest to the authors include the results of statistical analysis on the graphs.

Reviewer 4 Report

Dear Authors,

The manuscript entitled " Assessment of the effect of A-PRF application during the surgical extraction of third molars on healing and the concentration of C-reactive protein." However, major revisions are required in order the manuscript to be further processed. Below you can find my recommendations.

Major Recommendations

1) Introduction. The indroduction section should be more comprehensive and focused on the main topic of the current manuscript. Also, the introduction section should be shorten. E.g. lines 51-58, the presented growth factors should be presented in a strainght line accompanied by the description of each factor.

2) Materials and Methods, 2.1 Experimental Groups, lines 132-137. This section should be removed or replaced (e.g. the 60 patients were radomly distributed between control and study groups). Also, the Pic. 1 should be removed.

3) Materials and Methods. Line 169, Figure 2. There is no any mention of the figure in the main text and also no legend in the figure. In my opinion it is better, also a scale bar should be added in the figure 2.

4) Materials and Methods-2.8 Statistical Analysis. The authors should describe the Statistical software that used in this study.

5) Results -3.1 CRP. A short explanatory paragraph should be added. Also table II must be removed and provided as supplementary material. 

It could be better a bar chart to be added in this section, described the mean ± SD to be added. Also, the last line of table III describing the CRP Δ might be confusing for the readers, and it could be better to removed, and provided as supplementary material. 

In table III, CRPI study and Control group and CRP II control group, the SD has a wide range. Therefore, initial the authors have to validate again their results, and second the statistical analysis should be performed by a professional statistician. Maybe some outlier values need to be removed, in order to improve the data. 

In table III, the presented p test should be reduced to 3 digits only.

Table III, minimum and maximum values should be removed and presented only as supplementary material.

Pic 3. The legend should be in the bottom of the figure and not in the top of it. Also, in the presented diagramm the SD should be added.

In table V, same as Table III, regarding the p test, minimum and maximum. 

In addition, the WBC, RBC, PLT number before and the production of PRF should be presented in the main text.

Furthermore, the authors need to improve the used language in a more scientific way. Also, they must follow the guidelines for authors regarding the presented figures. e.g. Figure 1 instead of Pic 1.

What about using except CRP and additional inflammatory markers?

The manuscript needs major revision to all that comments.

Author Response

Thank you very much for valuable tips and comments, we are grateful for such an extensive review, which will serve us not only to improve the current manuscript, but also for the future.

Round 2

Reviewer 2 Report

The scientific level and content has been improved in the revision version. However, I still consider that the topic of the manuscript does not fit exactly  to the journal area. A minor adjustment of the references list will be wellcome, by adding some recent papers dealing with PRGF formulations,  Hematology Parameters (Leukocytes, Erythrocytes , Platelets , etc…) and Growth Factor Content such as Quantitative assessment of the main growth factors, cytokines, and chemokines, etc. I suggest the following: 

Ratiu et al. (2018) Appl. Sci. 2019, 9(5), 1035; https://doi.org/10.3390/app9051035, in which the complete characterization of the autologous formulation is presented.

Reviewer 4 Report

Dear Authors,

The majority of my comments have been well answered. However, i think that the bar chart on figure 3 needs further improvement. It is usefull to add besides the average and SD of the values.

In general, from my side the manuscript should further processed by the journal to the next step of the publication process.

Congratulations

Author Response

This manuscript is a resubmission of an earlier submission. The following is a list of the peer review reports and author responses from that submission.